# Targeted Nanocarrier Delivery of RNA Therapeutics to Control HIV Infection

**DOI:** 10.3390/pharmaceutics14071352

**Published:** 2022-06-26

**Authors:** Esinam E. Agbosu, Scott Ledger, Anthony D. Kelleher, Jing Wen, Chantelle L. Ahlenstiel

**Affiliations:** 1Kirby Institute, University of New South Wales, Sydney, NSW 2052, Australia; eagbosu@kirby.unsw.edu.au (E.E.A.); sledger@kirby.unsw.edu.au (S.L.); akelleher@kirby.unsw.edu.au (A.D.K.); 2RNA Institute, University of New South Wales, Sydney, NSW 2052, Australia; 3UCLA AIDS Institute, David Geffen School of Medicine, University of California Los Angeles, Los Angeles, CA 90095, USA; jingwen@mednet.ucla.edu

**Keywords:** RNA therapeutics, siRNA, nanocarrier, HIV, latent reservoir, delivery

## Abstract

Our understanding of HIV infection has greatly advanced since the discovery of the virus in 1983. Treatment options have improved the quality of life of people living with HIV/AIDS, turning it from a fatal disease into a chronic, manageable infection. Despite all this progress, a cure remains elusive. A major barrier to attaining an HIV cure is the presence of the latent viral reservoir, which is established early in infection and persists for the lifetime of the host, even during prolonged anti-viral therapy. Different cure strategies are currently being explored to eliminate or suppress this reservoir. Several studies have shown that a functional cure may be achieved by preventing infection and also inhibiting reactivation of the virus from the latent reservoir. Here, we briefly describe the main HIV cure strategies, focussing on the use of RNA therapeutics, including small interfering RNA (siRNA) to maintain HIV permanently in a state of super latency, and CRISPR gRNA to excise the latent reservoir. A challenge with progressing RNA therapeutics to the clinic is achieving effective delivery into the host cell. This review covers recent nanotechnological strategies for siRNA delivery using liposomes, N-acetylgalactosamine conjugation, inorganic nanoparticles and polymer-based nanocapsules. We further discuss the opportunities and challenges of those strategies for HIV treatment.

## 1. Introduction

The use of RNA interference (RNAi), a natural biological phenomenon within cells for silencing genes, in the treatment of tumours and rare genetic disease has rapidly progressed in the past few years. The process involves small interfering RNAs (siRNAs), which are noncoding and double stranded, or similar molecules, such as microRNAs, that mediate gene silencing. The specificity of targeting by siRNA is advantageous for treating any disease that requires genetic regulation. However, delivery of siRNA therapeutics to target cells in the body is a major barrier. The abundance of RNases in tissues results in rapid degradation of exogenous RNA. The size and highly negative charge of RNA molecules makes their delivery across cell membranes difficult [1]. To circumvent this challenge, major advances have been made in the development of RNA delivery vehicles, including viral and non-viral delivery systems. The use of viral delivery systems, such as lentiviral vectors and adeno-associated viral vectors, has been successfully applied in the efficient systemic delivery of RNAs. However, unwanted immune responses to viral vectors hinder their wide application in clinic [2,3]. Integration of the lentiviral vector close to or within a coding region of the host genome may also result in undesirable effects such as loss of gene function or overexpression, stimulation of oncogenesis, or cell death [4]. Therefore, non-viral systems have attracted increasing attention for their potential safety and flexibility. Non-viral vectors, such as nanocarriers, have high gene-packaging capacity, can be designed to have low immunogenicity and are relatively simple and less costly to manufacture [5]. Currently, four RNAi therapeutics delivered by non-viral strategies have been approved by the FDA and EMA for delivering RNAi into livers in clinics. Patisiran, the first approved RNAi (in 2018) for the treatment of hereditary transthyretin-mediated amyloidosis (hTTR) [6], acts by targeting the untranslated region of TTR mRNA, reducing translation of its protein [7]. The lipid nanoparticle formulation directs it to the main TTR production site, which is the liver (Figure 1). The second approved RNAi drug, Givosiran, has the siRNA molecule conjugated to Alnylam’s Enhanced Stabilization Chemistry N-acetylgalactosamine (ESC-GalNAc) residues, which binds to specific receptors mainly found on hepatocytes [8]. It is used in the treatment of acute hepatic porphyria (AHP) and acts by degrading 5-aminolevulinic acid synthase 1 (ALAS1) mRNA, resulting in reduced levels of intermediates associated with AHP attacks. The third approved RNAi drug, Lumasiran, was developed for the treatment of Hyperoxaluria type 1 and, similar to Givosiran, is conjugated to ESC-GalNAc. It degrades glycolate oxidase mRNA, reducing hepatic production of oxalate [9]. Inclisiran, the most recently approved drug, targets mRNA coding for hepatic proprotein convertase subtilisin-kexin type 9 and utilises the GalNAc tail. It results in a reduction of low-density lipoprotein cholesterol and is used for the treatment of atherosclerotic cardiovascular disease [10].

Viral infections, such as HIV infection, occur in various sites other than the liver. Therefore, RNAi therapeutics to treat viral infection require different non-viral strategies to direct them to the target cells. HIV primarily infects CD4+ T cells and macrophages, predominantly within the lymphoid tissues, and establishes latent reservoirs in multiple organs throughout the body [11], with some being sanctuary sites (Figure 2A). The proportion of infected cells that harbour latent viruses is small (one in a million), making them difficult to find and eliminate. Resting CD4+ T cells have low levels of endocytosis (and therefore low uptake of any therapeutic) and have proven to be difficult targets for the delivery of gene cargoes [12]. Due to this, it may be necessary to use targeted nanocarriers for the effective delivery of gene cargoes including RNA therapeutics. Approaches to target nanocarriers to specific sites are extensive; some of them are described in this review. For efficient control of HIV, therapy would need to be targeted at multiple cell types (involved in the latent reservoir) and multiple sites, as shown in Figure 2. Thus, a combination of different targeting approaches may be required. In addition, global infections are driven by highly diverse subtypes and strains of the virus, as a result of its dynamic evolutionary process. These factors, along with those indicated in Figure 2B, make HIV control challenging.

## 2. HIV Latent Reservoir

Antiretroviral therapy (ART) has turned an acute fatal infection into a manageable chronic disease, but it does not provide a cure. According to UNAIDS, there are ~39 million people living with HIV, with an estimated 28 million having access to ART [13]. ART is highly effective in inhibiting active virus replication. However, ART does not have any substantial impact on the integrated provirus, which remains transcriptionally inactive. It is estimated that, in fully ART suppressed individuals, latently infected cells are still in the range of millions amongst the total lymphocyte pool [14,15]. When ART is withdrawn, the virus rebounds to pre-therapy levels. Thus, to achieve a cure, effective strategies targeting latent reservoirs are critical. A major challenge to developing a cure is the persistence of HIV in latent reservoirs, such as resting CD4+ T cells and macrophages [16].

## 3. HIV Cure Strategies

Four main strategies specifically targeting the latent reservoirs have been attempted to date in the quest to obtain an HIV cure, including gene therapy by stem cell transplant, gene editing, and the “shock and kill” and “block and lock” approaches. In addition to the four main approaches that are discussed below, HIV-specific immunotoxins [17,18] and broadly neutralizing antibodies [19,20] have been explored for viral suppression and elimination of the latent reservoir.

### 3.1. Hematopoietic Stem Cell Transplant

Stem cell transplantation to clear HIV is invasive and highly challenging, however, the advantage over other treatments is that the HIV provirus or genes necessary for HIV expression or infection can be directly gene-modified without the need for lifetime therapy. The approach most advanced in preclinical and clinical studies is targeting the HIV-1 co-receptor, CCR5, via naturally mutated or engineered hematopoietic stem cells (HSCs) and rendering the cells resistant to HIV-1 infection after gene-modified HSC transplant. Currently, three patients have undetectable HIV using this approach. The first involved the use of HSCs that were inherently resistant to HIV infection, as in the “Berlin patient”. Donor cells were homozygous for a CCR5 delta-32 deletion, so in essence, they lacked an essential HIV entry co-receptor CCR5 and were resistant to R5-tropic infection. The patient underwent double allogenic haematopoietic stem-cell transplantation (HSCT) for the treatment of leukaemia, receiving total body irradiation with each procedure. No viral rebound was detected following discontinuation of ART after the transplant [21]. The procedure used in the second documented case of HIV-1 remission was less aggressive. The patient, known as the “London patient”, was diagnosed with Hodgkin’s lymphoma and underwent a single allogenic HSCT with CCR5Δ32/Δ32 donor cells and has been in HIV remission for over 30 months [22]. While the previous cases were men, the most recent case involved a woman of mixed race diagnosed with acute myeloid leukaemia. She received a transplant with a combination of CCR5Δ32/Δ32 umbilical cord blood HSCs and adult HSCs from a relative’s blood. With the exception of transient detection of trace amounts of HIV DNA in her peripherical blood mononuclear cells (PBMCs) 14 weeks after ART withdrawal, no HIV DNA has been detected for 14 months [23]. However, this approach is not likely to be a feasible mainstream cure for HIV because of its high cost, the relatively low numbers of CCR5 delta-32 donors who also need to be HLA-matched to recipients, the risk associated with bone marrow transplants, and naturally resistant HIV X4 strains which do not use CCR5 receptors for entry.

### 3.2. Gene-Editing of HIV Provirus

The gene-editing approach employs the use of nucleases to modify target genes. A potential outcome of this approach is excision of the HIV provirus from the host genome in latently infected cells. Efficient gene-modification activity has been achieved by a number of systems including clustered regularly interspaced short palindromic repeats (CRISPR)-associated protein 9 (CRISPR/Cas9) technologies [24], the transcription-activator-like effector nucleases (TALENs), zinc finger nucleases [25,26], and homing endonucleases [27]. Compared to other nucleases, the CRISPR-Cas9 system has gained increasing attention due to its simplicity. CRISPRs, first detected in *Escherichia coli*, were thought to be the immune system for bacteria [28]. CRISPR elements are located next to multiple conserved genes known as CRISPR-associated (Cas) genes. Several Cas systems have been identified as effectors of CRISPR in different organisms. The most commonly used effector protein is the *Streptococcus pyogenes* Cas9 endonuclease. The CRISPR-Cas9 system has been the preferred choice for genome editing, with most approaches based on it or its modification. Cas9 recognises a 5′NGG 3′ protospacer-adjacent motif (PAM) immediately adjacent to the target sequence [29]. A specificity-determining CRISPR RNA (crRNA) and a trans-activating RNA (tracrRNA) are required for target sequence recognition. The two RNA components can be fused to form a single guide RNA (gRNA) which complexes with the Cas9 protein and guides it to the target site [29]. The complex cleaves the target site to generate a double-stranded DNA break (DSB). The DSB can then be repaired using nonhomologous end-joining (NHEJ) or homology-directed repair (HDR) (Figure 3). NHEJ may result in random insertions and deletions at the site of cleavage and a loss of gene function. HDR directs the precise insertion or replacement of a DNA template with sequences homologous to the DSB site [30].

The CRISPR-Cas9 system has shown promising results in in vitro and, to an extent, in vivo excision of the HIV provirus from the host genome [31]. The CRISPR-Cas9 system targeting the proviral genome has demonstrated effective clearance of replication-competent virus in both ART-treated humanized mice [32] and non-human primates (NHPs) [33]. HIV-1-infected humanized mice were sequentially treated with long-acting slow-effective release antiviral therapy (LASER ART) in the form of fatty-acid-modified prodrugs of existing ARTs and CRISPR-Cas9 delivered by Adeno-Associated Virus (AAV). A higher reduction in viral DNA copy numbers was observed in previously infected tissues in the dually treated group compared to those on only either of the treatments. However, there was viral rebound after ART withdrawal in most cases, except in two of the dually treated mice [32]. The capability of the CRISPR-Cas9 system for HIV cure has also been tested in a proof-of-concept study with SIV-infected NHPs [33]. Intravenous administration of an AAV-delivered CRISPR-Cas9 molecule targeted specifically at the viral gag and LTR regions to the SIV-infected NHPs resulted in viral DNA excision in blood and tissues [33]. This system can also be employed to edit the genes of essential cellular factors for the replication cycle of the virus, rendering the cells permanently resistant to HIV infection. In a recent study, primary CD4+ T cells showed reduced susceptibility to HIV-1 infection following a CRISPR-knockout of interferon-stimulated genes (ISG-15) compared to cells treated with non-targeting controls [34].

Despite the promise of the CRISPR-Cas9 system, some potential disadvantages have limited its widespread application. A study used a CRISPR-Cas9 gRNA that targets both 5′ and 3′ LTRs to excise HIV proviral genome integrated in 293T cells, J-Lat cells and human leukaemia cells and reported that the excised provirus persisted in cells for weeks, closing to form a circular element that may be reactivated under certain conditions [31]. Off-target effects have also been reported. As observed by Fu et al., CRISPR-Cas9 based RNA-guided nucleases tolerated single and double mismatches in human cells depending on their positions in the DNA-gRNA interface. The frequency of mutagenesis at off-target sites with up to five mismatches were comparable, or in some cases, higher than those that occurred at the target site. This poses a problem for therapeutic applications as these tools may introduce undesirable changes in unrelated genes [35]. Moreover, the large size of Cas9 nuclease (160 kDa) poses a delivery challenge in packing into viral vectors, and the limited NGG PAM sites for Cas9 to bind and introduce genomic mutations may be not sufficient to find an optimized targeting sequence with minimal off-target effects [36]. Therefore, there is an urgent need for alternative Cas systems with smaller-sized effector proteins, more flexible PAM requirements, and different substrate preferences. The re-engineering of Cas9 proteins is being explored to enhance their packaging and delivery, increase fidelity and widen the targeting scope [37]. More variants of Cas9 and other Cas nucleases have been discovered for gene editing in mammalian cells, including spCas9-NG, base editing, xCas9, Cas12a (Cpf1), Cas13 and Cas14 (a/b/c) [36]. An example of a recently developed system is based on modification of CRISPR-Cas13b, and was shown to potentially silence viral RNA in cultures [38]. Researchers were able to use the CRISPR-Cas9 system to reprogramme the effectors of the endonuclease, which supressed multiple variants of Severe Acute Respiratory Syndrome Coronavirus 2 (SARS-CoV-2) and prevented viral escape [39].

### 3.3. Shock and Kill

The “shock and kill” approach seeks to effectively “awaken” virus in the dormant cells and then allows those cells to die or be killed, and it is the most widely studied approach for achieving a cure. This strategy aims to eradicate the reservoir by two steps: (1) utilizing latency-reversing agents (LRAs) to activate cellular transcription factors for HIV expression [40], and (2) inducing recognition of the infected cells by the host immune system and subsequent cell-killing by either the natural immune response or various therapeutics, such as anti-HIV antibodies and engineered T cells, respectively, leading to a reduction of the latent reservoir size. Continuation of ART during viral reactivation has been suggested along with “kill” therapeutics to prevent infection with newly produced virus. Reactivation is achieved through engaging components of signal transduction pathways, transcription factors and cofactors. The main classes of LRAs include epigenetic modifiers, inhibitors or activators of cellular factors, and immunomodulators [41]. Epigenetic modifiers, such as Histone Deacetylase Inhibitors (HDACi) and histone methyltransferase inhibitors (HMTi), inhibit the activity of enzymes involved in the post-translational modifications of histones that maintain HIV latency [42]. HDACi are the most studied class of LRAs, with several drugs studied in clinical trials, such as valproate, Panobinostat, Vorinostat and Romidepsin [43]. Although some increments in the levels of cell-associated unspliced HIV-RNA have been observed, they have not resulted in a reduction in the size of the viral reservoir [43]. Besides epigenetic modifiers, key factors in the signal transduction pathway, such as Protein kinase C, which induces signalling via the transcription factor NF-κB, have also been tested for reversing HIV latency, including PKC agonists, Bryostatin, Prostratin and Ingenol [40]. Moreover, toll-like receptors, particularly 7, 8 and 9, are being studied as LRAs to stimulate targeting and immune clearance of the reactivated cells. These approaches on their own have not been very successful in vivo to date [44]. A potentially successful approach would be to utilize a combination of these classes of LRAs [40]. This needs to be balanced with potential toxicities to ensure safety. The “kill” step is also a major limiting factor in this approach and needs to be extensively developed to advance this cure approach.

### 3.4. Block and Lock

The final approach to an HIV cure is known as “block and lock”. In contrast to the three previously mentioned approaches to eliminating HIV, the “block and lock” approach aims to provide a functional cure, which maintains the virus in a permanently latent state by targeting key steps in the transcriptional machinery [42]. This would prevent viral replication, production of new virions and subsequent damage to the immune system. This can be achieved by introducing small molecules to inhibit viral and cellular proteins involved in activation of the transcription pathway. Currently, the most studied therapeutic of “block and lock” is a Tat-inhibitor, didehydro-cortistatin A (dCA) [45]. Tat protein is a critical factor for the elongation of HIV transcripts, as it interacts with and recruits the complexes required by RNA polymerase II for transcriptional elongation. Studies with dCA have shown significant inhibition of virus replication and resistance to viral reactivation upon administration of LRAs [46]. HIV requires its integrase (IN) for integration into the host genome. Interactions between IN and the cellular chromatin-tethering factor LEDGF/p75 direct specific integration into active transcription sites. Small molecules (LEDGINs) have been designed that inhibit this interaction, with promising results [47,48]. Small interfering ribonucleic acids (siRNA) are another promising alternative approach with high specificity to maintain repression at the HIV promoter region [45,49]. Repression of the HIV promoter prevents transcription and blocks formation of new virions, known as transcriptional gene silencing (TGS) (Figure 4). Such siRNAs can also be designed to target regions within viral RNAs or host mRNAs that are highly conserved and are essential for the viral replication cycle and trigger their targeted cleavage, known as post-transcriptional gene silencing (PTGS) [50] (Figure 4). The transcriptional gene silencing or “block and lock” approach aims to tightly control viraemic levels and to provide a functional cure similar to what is observed in elite controllers.

## 4. RNA Interference (RNAi) Therapeutics for HIV Treatment

RNA therapeutics have been widely explored for HIV treatment, including siRNAs to maintain HIV permanently in a state of super latency (the block and lock approach), shRNAs targeting CCR5 mRNA, and CRISPR gRNA to excise the latent reservoirs. As mentioned above, there are two main RNA-induced gene-silencing pathways, PTGS and TGS.

Both pathways utilise sequence-specific silencing triggered by double-stranded RNA, usually 19–22 bp long [51]. The dsRNAs are loaded on Argonaute proteins (AGO) and target by complementary base-pairing. The targets are either mRNA transcripts or gene promoters. The two pathways differ in their target, the location within the cell, and the type of Argonaute proteins mediating the silencing. PTGS is well described, understood and exploited. It predominantly occurs in the cytoplasm and results in direct cleavage of the targeted mRNA [52]. Double-stranded RNA is fragmented by Dicer and is loaded onto the Argonaute-2 protein (AGO2) to form the pre-RNA-inducing silencing complex (RISC). There is cleavage and removal of the passenger strand, activating the mature RISC. The remaining RNA strand serves as the guide RNA within the RISC, which binds to complementary regions of mRNA, resulting in cleavage of the mRNA or repression of its translation [52].

TGS on the other hand is less understood. Small noncoding RNAs (sncRNA) generated and processed into miRNAs in the nucleus or exogenous siRNAs are loaded onto the AGO1-associated RNA-induced initiator of the transcriptional gene silencing (RITS) complex. The RITS complex interacts with epigenetic remodelling proteins, recruiting them to target loci homologous to the mi/siRNA in the genome, resulting in histone and DNA methylation and ultimately localized chromatin compaction and epigenetic silencing [53,54].

The high sequence specificity of base-pairing in PTGS reduces the chances of off-target silencing. However, a single mutation in the target sequence would reduce gene silencing. The silencing effects of PTGS are temporal, as it leads to the degradation of already transcribed mRNA, but does not affect transcription of the gene. Thus, therapeutics based on PTGS require continuous delivery. The epigenetic modifications to gene expression resulting from TGS appear to be more stable, while it is also heritable and potentially long-lasting [53], decreasing the need for sustained delivery.

Several RNAi approaches have been investigated over the years for the treatment of HIV-1. They are mainly grouped into two approaches, based on the mode of action: RNA targeting via antisense-based mechanisms and protein targeting via decoy or aptamers [55]. RNA molecules (decoys) that mimic the structures of transactivation response (TAR) and Rev response element (RRE) in HIV-1 RNA are used to inhibit HIV-1 Tat and Rev proteins [56]. Single-stranded anti-sense RNA (ASOs) [57], ribozymes [58] and shRNAs [59] through complementary base pairing to the target RNA sequence have also been extensively researched for inhibiting RNA translation, binding to and cleaving target RNA sequences, or for recruiting ribonucleases to degrade the target RNA. The different approaches can be used individually or in combination. An example of a combinatory approach was employed in a clinical trial carried out in patients undergoing transplantation for AIDS-related lymphoma using CD34(+) cells modified with a lentiviral vector encoding shRNA against tat/rev, ribozyme against CCR5 and TAR-decoy targeted at Tat [60]. Many RNAi therapies for HIV treatment have been targeted at CCR5, usually in combination with other targets [59,60]. The most advanced RNAi therapeutic designed to downregulate CCR5 expression is the Cal-1 (LVsh5/C46) drug product. It comprises a lentiviral vector encoding both for an shRNA for downregulation of CCR5 and a fusion inhibitor, C46. Preliminary studies showed the drug to be nontoxic and effective at preventing infection with HIV-1 in vector-transduced cells. The drug is currently in phase I/II trials [61,62].

## 5. Non-Viral Delivery Systems

A major challenge with the use of RNA therapy is the mode of its delivery into the target host cells, particularly in the case of HIV where the latent reservoir is distributed throughout the body. RNAs are rapidly cleared from circulation, not easily taken up by cells, and are inefficient at endosome escape in target cells [63]. A delivery vehicle is required to circumvent these barriers. The choice of delivery system is based on the efficiency of its immunogenicity, ability to avoid renal clearance, uptake by the target cell, and specificity [64]. Lentiviral vectors (LVs) are efficient and commonly used tools for gene transfer, although they are associated with some risks, particularly the uncontrolled nature of the integration of transgenes into sites in the host genome, which may cause oncogenesis. Low-density lipoprotein receptor, the receptor for the commonly used vesicular stomatitis virus (VSVg) envelope in LVs, is not typically present in resting CD4+ T cells, necessitating the use of a different envelope. The native HIV envelope may be used in some cases, which also increases the risk in LV use [12]. However, research is ongoing to develop LVs that drive site-specific integration. In one study, the integration machinery was modified to home to and target highly repetitive regions predominantly localized in ribosomal DNA. No observed DNA deletions or negative impacts on cell health were associated with integration in these sites in transduced primary human T cells [65]. AAV is also widely used for RNAi delivery. Despite their effectiveness, viral vectors have been severely hampered by unwanted host immune responses, associated oncogenicity (due to uncontrolled integration) and expensive production costs [3,66]. An alternative to viral vector delivery is non-viral vector delivery.

### 5.1. Nanocarriers

Nanocarriers (NC) are generally defined as particles with a size between 1 and 100 nm, although the term can also be used to describe particles with a size of up to several hundred nanometers [67]. Depending on their size and surface properties, nanocarriers have distinct qualities [68] (Figure 5). Their relatively small sizes and tuneable surface properties make them an attractive choice for medical use. They are able to bind to, adsorb and transport other molecules like proteins, drugs and probes because of their larger surface area [69]. Another advantage they offer over other carriers is that their physiochemical properties can be optimized to enhance their uptake by cells, stability in the blood, and immune clearance. For these reasons, nanocarriers are currently being investigated as delivery platforms for RNA therapeutics targeting HIV.

The size, shape, and surface properties of nanocarriers affect their circulation and distribution in the body. One limiting factor for nanocarriers is poor biodistribution, with the liver, spleen, and kidney prevailing as the major targets of passive delivery (i.e., they do not include a specific targeting moiety) [70]. The appropriate size of the particle reduces the clearance rate and increases circulation time in the bloodstream. This increases bioavailability and allows for accumulation of the nanocarriers at the target site. Smaller particles (less than 10 nm) are easily filtered by the kidney while larger ones (greater than 200 nm) are cleared by the phagocytic cells [71]. The surface charge of nanocarriers also plays a key role in their distribution. For example, the positively charged nanocarriers accumulate rapidly within the liver and spleen [72]. Neutral nanocarriers show longer life in circulation and are effective for therapeutic delivery [73,74]. Moreover, because nanocarriers used for drug delivery are internalized by the targeted cells, their shape is also important for biodistribution. Rod-shaped cationic nanocarriers, for example, are simpler to absorb by endosomes than other cationic nanocarrier shapes, implying that immune cells may recognize these nanocarriers as rod-shaped bacteria [75]. A summary of current siRNA therapeutics utilising nanoparticle systems for delivery are listed in Table 1.

#### 5.1.1. Lipid Nanoparticle

One of the most successful nanocarriers for siRNA delivery is the lipid nanoparticle (LNP). In comparison to viral vectors, lipid-based NPs generally have a lower toxicity and immunogenicity, are more structurally flexible and biocompatible. There is also a relative ease in large-scale production of LNPs [103]. They are made up of ionizable cationic lipids which encapsulate the RNA molecule. Two major classes of LNPs, liposomes and lipid-like nanoparticles, are commonly used in passive delivery of nucleic acid [104]. The first approved siRNA therapeutic, Patisiran, was formulated using this platform [6]. The Moderna vaccine, mRNA-1273, and the Pfizer-BioNTech vaccine, BNT162b2, against SARS-CoV-2 are the most recent approved therapeutics based on the lipid nanoparticle system [105].

Liposomes consist of phospholipids with a hydrophilic head and two hydrophobic chains which form a bilayer around an aqueous core (Figure 5). Lipids such as 1,2-dioleoyloxy-3-trimethylammonium propane chloride (DOTAP) and N-[1-(2,3-dioleoyloxy)propyl]-N,N,N-trimethylammonium chloride (DOTMA) that are classically cationic were previously used in the delivery of nucleic acids [63]. The electrostatic interactions between positively charged lipids and negatively charged siRNA form the basis of the liposome-nucleic acid complex. Though effective, cytotoxicity associated with cationic lipids has emphasised the need for using lipids that are ionic only under certain conditions. Liposomes made of ionizable lipids that are neutral at a physiologic pH, but become protonated and charged under acidic pH conditions, such as in endosomes and lysosomes, are currently being used in the formulation of lysosomes in order to overcome this cytotoxicity. An example of such a lipid is DLin-KC2-DMA (1,2-dilinoleyl-4-(2-dimethylaminoethyl)-[1,3]-dioxolane) [106].

#### 5.1.2. Inorganic Nanocarriers

Because of their advantages of precise size control, variable surface characteristics, and high drug loading efficiency, inorganic nanoparticles (NPs) are emerging as a desirable nanocarrier for RNA delivery. Inorganic molecules exploited for nanoparticle formulations include metals, magnetics, quantum dots, and calcium phosphate. Gold (Au) is one of the preferred metals used for NPs (shown in Figure 5). It is less toxic and has low immunogenicity. The surface of Au NPs can easily be modified with different compounds and ligands to increase efficiency and targeting [63]. RNA fragments can be conjugated to the surface of the Au particle, and this RNA–Au conjugate has been observed to cross cell membranes. The conjugation is done either by using thiol or via electrostatic interactions, with the stronger bond form in the thiol–Au interaction [107]. To increase the cellular uptake, the AuNP can be coated with another cellular delivery agent. In one study, siRNAs were conjugated to AuNP modified with polyethylene glycol. The siRNA–Au NP were then coated with end-modified poly (β-amino ester)s (PBAEs). PBAEs are cationic polymers which compact nucleic acids and enhance their cellular uptake. Increased levels of siRNA delivery were observed in vitro [108]. Au NPs have successfully been used to deliver epidermal growth factor receptor (EGFR)-targeted siRNA therapy topically through the skin. The constructs permeated the skin of mice, hindered the expression of EGFR and reduced the thickness of the epidermis in treated cells [109].

#### 5.1.3. Polymer-Based Systems

Polymer-based systems have been widely studied and employed among the different approaches for siRNA delivery. Varied formulations can be synthesised, each with different physicochemical properties, to suit various applications. Derivatives of the initial formulation can be easily created by substitution, addition and removal of functional groups to enhance the desired characteristics or eliminate structural and functional limitations [110]. This makes them adaptable and useful for the delivery of siRNAs. The system may be based on natural materials and their derivatives or on chemically synthesised molecules. The polymers may be directly conjugated to the siRNA, or a complex block or shell may be formed with or around the siRNA molecules to protect them from nucleases and rapid renal clearance.

siRNA-polymer bioconjugates

Polymer-based systems can be developed using naturally occurring materials. These natural materials include oligo- and polysaccharides, lipids and cell-penetrating peptides (CPP). An earlier example of this is the use of cyclodextrin in a self-assembled NP formulation known as CALAA-01 [111]. The formulation, which consisted of cyclodextrin-containing polymer (CDP), a polyethylene glycol (PEG) stabilisation agent, and human transferrin to enhance targeting of transferrin-receptor-expressing tumour cells, was intravenously administered to cancer patients. Significant reductions in target mRNA levels and protein product were observed [111].

Cell uptake is a major challenge in siRNA therapeutics, however, conjugation of siRNAs to peptides, particularly CPPs, can be used to overcome this. Proteins such as poly-L-lysine (PLL) have been widely studied for nucleic acid delivery due to their biocompatibility and biodegradability [112]. A drawback of PLL-siRNA complexes is their likelihood of binding to serum proteins, which reduces the silencing efficiency. This can be mitigated by addition of a stabilising molecule [113]. Synthetic proteins have also been generated for the purpose of nucleic acid delivery. A study by Gabrielson et al. demonstrated the efficiency of a synthetic helical polypeptide, PVBLG_n_-8, to non-specifically destabilise cell membranes and deliver siRNA to cells [114].

Polymeric complexes

Two commonly used polymers in the medical field that have been explored for siRNA delivery are poly (ethylene glycol) (PEG) and poly (D,L-lactic-co-glycolic acid) (PLGA). PLGA is a biocompatible hydrophobic polymer which can be conjugated to hydrophilic molecules. When directly conjugated to a siRNA, the hydrophobic PLGA units aggregate to the core, with the hydrophilic units of the siRNA surrounding it to form micelles. Addition of a cationic polymer like polyethylenimine (PEI) increases stability of the complex [115]. PEG–siRNA conjugates have demonstrated a slower clearance rate and broader tissue and organ distribution. Gene silencing, however, is reduced compared to unconjugated molecules. This is attributed to the steric hindrance created by the PEG molecule, which hinders docking of the siRNA into RISC [116]. This hindrance can be overcome by incorporating a linker which holds together the complex during circulation but releases the siRNA inside a cell. Steric hindrance of polymers can, however, be used to control siRNA activity. This technique has been employed in the formulation of hydrogels for delivery of siRNA to cartilaginous tissues. Hybrid hydrogels were composed of poly(N-isopropylacrylamide) (pNIPAAM) and layered double hydroxides (LDHs). In addition to being large, pNIPAAM is randomly coiled below its lower critical solution temperature (LCST), blocking the siRNA from docking into RISC. When in an environment with a temperature higher than its LCST, it stacks to form a compact globule which allows the siRNA to access the RISC [117].

To enhance the efficacy of an siRNA therapeutic, multiple factors such as its immunogenicity, ability to avoid renal clearance, uptake by the target cell, and specificity have to be taken into consideration [64]. In many cases, one vector or strategy may be insufficient for overcoming all hurdles. Therefore, it is beneficial to combine different polymer molecules with different functions to create a multiconjugate system. The first such system was developed by Roxema et al. and was termed dynamic polyconjugate (DPC) [118]. The siRNA molecule was attached to the amphipathic poly (vinyl ether) (PBAVE) backbone through a disulphide linkage. PEG chains for shielding and N-acetylgalactosamine ligand to target hepatocytes were attached to PBAVE using a pH-sensitive linker. Further, siRNA was specifically delivered to hepatocytes and in acidic environments such as the endosome, the pH-sensitive linker was cleaved, releasing the siRNA [118]. Several polyconjugates have since been developed with different backbones.

A novel method of polymeric delivery of siRNA has been described by Yan et al. This method involves the formation of a polymer network around the surface of a single molecule of siRNA, making it a polymer NP (Figure 5). This greatly increases stability and enables effective endosomal escape, achieving an enhanced release of siRNA into the cytoplasm [119]. The nanocapsule system is flexible, allowing for relative ease in its modification to increase affinity, reduce immunogenicity, prolong circulation time and control the release of the content. A specific example of this system utilises acryl-spermine and tris-acrylamide as the polymers, and degradable crosslinker (Glycerol 1,3-diglycerolate diacrylate, GDGDA) and non-degradable crosslinker (N,N′-methylenesbisacrylamide, BIS) to conjugate the polymers to the surface of the siRNA molecule [96]. Yan et al. (2015) encapsulated a DNA cassette for shRNA for knocking down CCR5 in nanocapsules, which were engineered for timed release of the DNA cargo. They observed prolonged downregulation of CCR5, which indicates prolonged protection from HIV-1 infection. The system was also used to deliver microRNA-125b transiently to CD34+ cells, which resulted in a marked reduction of apoptosis and enhanced survival of CD34+ cells in culture.

For specificity of release of the content of the polymer, crosslinkers that are sensitive to particular conditions at the target site or within the target cell can be incorporated. In this case, peptide crosslinkers sensitive to cleavage by the HIV-1 protease were used. Only within cells that expressed functional HIV-1 protease would the crosslinker be cleaved and the polymer shell broken to release its contents. The polymer shell, created with positively charged N-(3-aminopropyl) methacrylamide, neutral N-(3-aminopropyl) methacrylamide and with a bisacryloylated peptide crosslinker, was made to encapsulate the toxin Ricin A. The release of Ricin A only within target cells in vitro resulted in the killing of HIV-1-producing cells without a cytotoxic effect on non-target cells [120].

In another application of the nanocapsule system, therapeutic monoclonal antibodies were used to successfully target the delivery of their cargo in vivo to metastases in the central nervous system (CNS) of murine models of non-Hodgkin lymphoma. Polymer shells, formed using neutral monomers with zwitterionic properties, with 2-methacryloyloxyethyl phosphorylcholine (MPC) as the monomer and glycerol dimethacrylate (GDMA) as the crosslinker, as well as ammonium persulfate and tetramethylethylenediamine as the initiator, were encapsulated around the anti CD20 antibody Rituximab. CXCL13, a ligand for the receptor CXCR5, was conjugated on the surface of the polymer shell [121]. CXCR5 is frequently expressed on B-cell lymphoma, allowing for targeted delivery of the nanocapsulated antibody to CXCR5-expressing lymphoma cells. This resulted in increased levels of the drug in the CNS and control of metastases in the mice. This system could serve as a model for delivering therapeutics with limited CNS penetration [121].

A challenge in the treatment of infections of the CNS is poor penetration of the blood–brain barrier (BBB). Modification of the polymer shell to contain abundant choline and acetylcholine analogues has been shown to deliver macromolecules to the CNS with >10-fold higher levels compared to native forms. This modification was utilised in treatment of HIV infection in the CNS of rhesus macaques. IgG1 antibody PGT121 was encapsulated via in situ polymerization of MPC as a monomer, Poly(DL-lactide)-b-Poly(ethylene glycol)-b-Poly(DL-lactide)-diacrylate triblock (A102, PLA-PEG-PLA) as a hydrolysable crosslinker, and GDMA as a degradable crosslinker. Effective transport of the nanocapsule-antibody across the BBB and into the CNS was observed [122].

#### 5.1.4. N-Acetylgalactosamine Conjugation

The host cellular defences recognise double-stranded siRNA as invading material. To circumvent this, several modifications to the backbone of the RNA molecule have been investigated. Modifications to the 2′ position of the ribose sugar have been shown to increase the stability of siRNA. This has been exploited to directly conjugate targeting domains to siRNAs, without requiring LNPs [123], making the formulation much less bulky. The most well-studied and commonly used siRNA conjugate is the trimer of N-acetylgalactosamine (GalNAc) (Figure 5). GalNAc binds to the binding protein asialoglycoprotein (ASGPR), expressed predominantly on hepatocytes with a very strong affinity [124], making this an easy way of actively targeting siRNAs to the hepatocytes. This is desirable, as GalNAc-siRNA conjugates are relatively simpler to synthesise. After binding to ASGPR, the GalNAc-siRNA is internalised into clathrin-coated endosomes and is released from the binding when the endosome becomes acidic. Glycosidases within the endosome cleave GalNAc from the siRNA. Although most of the free siRNA stays in the endosome, small amounts get transported into the cytoplasm to induce an RNAi response [123]. Three biotech companies are spearheading the development of therapeutics utilising GalNAc–siRNA conjugates, each making their unique modifications to the structure. Modifications by Alnylam Pharmaceuticals have included replacing all the 2′-OHs with an alternating 2′-F and 2′-OMe pattern in their standard template and subsequently decreasing the 2′-F while increasing the 2′-OMe content [125]. The Arrowhead Pharmaceuticals platform has utilised a combination therapy of cholesterol–siRNA conjugate with a peptide-GalNAc-carboxy dimethyl maleic anhydride linkage conjugate [126]. Dicerna Pharmaceuticals utilises a tetra-antennary GalNAc-dicer substrate siRNA conjugate [123]. Examples of RNA therapeutics utilising these platforms in clinical trials are listed in Table 1.

## 6. Surface Modification of Nanocarriers to Improve Delivery Efficiency

RNA therapeutics have far-ranging applications for many diseases that would be curable by gene silencing. The choice of the type of nanocarrier to use in delivering an RNA therapeutic may be determined by several factors, such as the disease, the target cell type (its size, structure of cell membrane-permeability, receptors), and the location of the cells. The effect of RNA therapeutics has been explored for treating various diseases in pre-clinics or clinics, targeting different cells in varied organs (Table 2). All nanocarriers currently in trials are lipid-based with no specific moiety, which results in passive targeting and accumulation in the liver. This may be desirable and relatively straightforward for diseases that affect the liver, as is the case in Hepatitis B infections and the currently approved siRNA therapeutics. However, to achieve a wider tissue biodistribution for HIV treatment, nanocarriers must be designed to minimize accumulation in the liver, avoid clearance by the immune system, and achieve effective, active delivery to the target tissues [127].

Targeting a single body system, such as the respiratory tract, has been investigated for developing RNA therapeutics for SARS-CoV-2, which infects the upper and lower respiratory tract. SARS-CoV-2 initially infects the airway and alveolar epithelial cells, vascular endothelial cells, and alveolar macrophages using the angiotensin-converting enzyme 2 (ACE2) receptor [128,129]. Candidate drugs developed exert therapeutic effects by preventing virus entry into cells, inhibiting viral enzyme activities, reducing virus-induced inflammation, and balancing immunomodulatory effects in the host [130]. Nanocarriers provide an opportunity to exploit the nasal delivery of gene therapy or other therapy against diseases directly to the respiratory tract. To achieve this, the nanocarriers have been administered through the inhalation route. This resulted in achieving pharmacologically effective drug concentrations and reducing possible systemic adverse side effects [131]. Additionally, nanocarriers were designed to overcome several physiological barriers, such as airway mucus, and to be small enough to be taken up by the cells in order to deliver desirable pharmacological effects [132].

The level of complexity for targeting a disease including more than one body system is exemplified by cystic fibrosis, which is an autosomal recessive disease caused by defects in the cystic fibrosis transmembrane conductance regulator (CFTR) protein in mucus- and sweat-producing cells, resulting in a thick and sticky mucus blocking the pathways [133]. It primarily affects the lungs and the gastrointestinal tract. Lung disease is however the predominant cause of death in people with the condition. Gene therapy to correct the abnormalities of the CFTR protein are being investigated [88,134,135].

The ultimate challenge for targeted nanocarrier delivery is found in HIV infection, which is multi-systemic and may require more than one active targeted approach for therapeutic delivery to all required cells. As mentioned in previous sections, HIV primarily infects CD4+ T cells and macrophages in multiple organs throughout the body, with the lymphoid tissues being the predominant sites [11]. Latent HIV-1 reservoirs are established in transcriptionally silent cells. Research into the actual cell types and the size of the latent reservoir is still ongoing. However, resting CD4+ T cells and memory T cells are thought to be the major cellular reservoirs [16]. They do not produce viral proteins and as such do not trigger cell death and are also not recognised and cleared by the immune system. As mentioned earlier, these cells have low rates of endocytosis, which hinders the uptake of therapeutics and poses a challenge for gene therapy. The spleen, thymus, lymph nodes, gut-associated lymphoid tissues and the bone marrow are the most important sites. The reservoirs can also be found in organ-specific cells such as astrocytes, epithelial cells, microglia and podocytes in the brain, lungs, the central nervous system and kidneys [11]. An effective delivery of therapeutics in such a multi-systemic way requires a route of administration that would allow bypass of the first-pass liver metabolism. Nanotherapeutics targeted at the latent reservoir would be most beneficial if administered by intravenous (IV) injection, which will deliver them into the circulatory system for distribution to the different organs. Modification to the nanocarrier to include a cocktail of targeting moieties that recognise patterns on the different cell types involved will be necessary to increase the efficacy of RNA therapeutics for HIV treatment. An important point to consider is the potential formation of protein coronas following IV administration. Protein coronas (PC) are formed by assembly of host proteins and opsonins around synthetic material, such as NCs, when introduced into the blood. The composition of the PC is influenced by factors including type of material, size and surface charge, and may affect the stability, size, biodistribution, rate of clearance and interactions of the NCs [136]. The PCs may also mask the targeting ligands, affecting the NC’s targeting ability and efficiency. Strategies have been adapted to enhance the stealth properties of NCs. These include incorporating a coat of synthetic and “self” molecules to limit interactions with host proteins and increase immune escape [136,137].

### 6.1. Targeting Moieties for Specific Cell Types

Actively targeting delivery make it possible to raise the RNA concentration at the target location and frequently facilitates internalization, resulting in improved efficacy and minimized off-target effects. Active targeted delivery of nanocarriers can be achieved through their conjugation with targeting moieties, which have an affinity to specific cellular molecules on the target cells [138]. The targeting moieties include peptides, aptamers and antibodies (Table 3) [139].

Antibodies have been extensively studied and used in several therapeutic systems [140]. They recognise specific antigens present on the surface of target cells. Antibodies have a high specificity and affinity for the target of interest as a result of the presence of two epitope binding sites in a single molecule. Monoclonal antibodies have been widely utilised for such purposes. This approach has been used to site-specifically deliver therapy to tumour cells over-expressing HER2 receptors. Herceptin, an antibody with affinity for HER2, was conjugated to polymeric micelles [141]. In another cancer therapy, single chain prostate stem cell antigen antibodies (scAbPSCA) were used to actively target polymer vesicles containing docetaxel to tumours [142]. Transferrin and bradykinin B2 antibodies have also been used to target chitosan-siRNA nanoparticles across the BBB to astrocytes [95]. Therapeutic antibodies are usually from murine sources and may potentially be immunogenic in humans. Another challenge is the bulkiness of the antibody and the cost of production. To counteract this, advances in antibody engineering have led to the development of antibodies that are humanized, chimeric or fragmented [140]. Additionally, endosomal escape after internalisation of the receptor-NC complex may be a challenge. Depending on the mechanism of endocytosis, the NCs may enter a degradative pathway [137]. In the case of HIV, antibodies to CD4+ T cell subsets and perivascular macrophages, representing the brain reservoir, are currently being explored in order to increase therapeutic delivery.

Peptides are thought to be a promising targeting moiety for delivering nucleic acids. They are smaller, stable, less immunogenic and when compared to antibodies, relatively less complex to produce [143]. Peptide-targeting ligands are identified by screening through libraries. The ligands used typically range in length from 10 to 15 amino acids. Peptides can selectively bind to receptors on tumour cells, for example, and disrupt ligand–receptor interactions on such cells, leading to the inhibition of cellular proliferation [144,145]. An example of peptide-targeting is the use of Cilengitide, a cyclic peptide with a binding affinity for integrins, for the treatment of non-small cell lung cancer and pancreatic cancer [146]. In a similar manner, Mucosal Vascular Addressin Cell Adhesion Molecule 1 (MAdCAM-1) has been used to deliver siRNA-bound LNPs specifically to leukocytes homing to intestinal tissues for the treatment of inflammatory bowel disease [147].

Aptamers are small nucleic acids that fold to form peculiar conformations with ligand-binding ability. The binding that occurs is highly specific and sensitive. This specificity together with their small size, low immunogenicity and the ability to easily penetrate target cells makes aptamers advantageous over antibodies [148]. Aptamers are identified and prepared through an in vitro chemical process called systemic evolution of ligands by exponential enrichment (SELEX), which can produce large batches with minimal variations in binding affinity [149]. Pegaptanib was the first aptamer to be approved for human use. It binds to and inhibits vascular endothelial growth factor (VEGF) 165 and is used in the treatment of ocular vascular disease [150]. A further, well-characterized aptamer, 2′-fluoro-pyridine-RNA, targets the prostate-specific membrane antigen (PSMA). PSMA is a transmembrane protein that is upregulated in many cancers. This aptamer has been used to deliver drug molecules and nanoparticle systems [151].

Small molecules are another group of targeting molecules with great potential. Their small size as well as ease and low cost of production makes them a viable therapeutic tool. Small molecules are recognised by and selectively bind to cell surface receptors, enhancing the delivery of the conjugated nanoparticle to the target cell [140]. The most-used small molecule for targeting is folic acid, which has a high affinity for folate receptors. Folate receptors are generally overexpressed in tumour cells, therefore, conjugating an anti-tumour therapeutic with folate results in targeted delivery of the drug to the tumour cells. Another common molecule used for targeting is GalNAc. With a high binding affinity for hepatocyte-specific asialoglycoprotein receptor (ASGR), conjugating this molecule to compounds involved in the treatment of liver-related diseases greatly increases the drug’s efficiency [152]. 

**Table 3 pharmaceutics-14-01352-t003:** Examples of targeting moieties currently used to increase drug delivery.

Type	Ligand	Targets	Description	Reference
Antibodies	Herceptin scAbP-SCA Anti-B2R	HER2Anti-prostate stem cell antigen Bradykinin B2 receptor	High binding affinity High cost of production	[95,141,142]
Peptides and proteins	Cilengitide MAdCAM-1 CXCL13	Integrins Integrin α4β7 + CXCR5 receptor	Low immunogenicity High binding affinity	[146,147]
Aptamers	2′-fluoro-pyridine-RNA aptamer Pegaptanib	Prostate specific membrane antigen VEGF receptor	High specificity and sensitivity Low immunogenicity Low molecular weight High cost of production	[150,151]
Small molecules	Folate GalNAc	Folate receptors Asialoglycoprotein receptor	Low molecular weight Ease of production	[140,152]

### 6.2. Approaches for Linking Targeting Moieties to NCs

Modifying the surface of NCs with ligands is achieved through three main methods (Figure 6). They involve adsorption of the ligand onto the surface of the NCs, the use of adapter molecules, and chemically conjugating the ligand to the NCs. Targeting moieties can be adsorbed onto the surfaces of NCs by utilizing the basic physicochemical properties of the molecules on the surface of NCs. Weak interactions, such as hydrophobic and electrostatic forces and hydrogen bonding, result in physical adsorption of the ligands to the NCs [153]. Ionic binding results from the attraction between opposite charges on the surface molecules of the NCs and the targeting ligand. These interactions are weaker and less stable than covalent bonds. Covalent conjugation is frequently used in linking targeting moieties to NCs, with the most commonly used method based on carbodiimide chemistry. NCs with carboxyl groups are treated with 1-ethyl-3-(-3-dimethylaminopropyl) carbodiimide (EDC) to activate their carbonyl moieties which then couple to the amino group on the targeting ligand to form a covalent amide bond [154]. Sulfhydryl groups on the side chains of ligands can be exploited for covalent conjugation. Free sulfhydryl groups react with maleimide groups to form a thioester linkage which is stable. Maleimide groups can be generated on the surface of NCs bearing primary amines and that are subsequently linked to sulfhydryl groups using maleimide cross-linking reagents. Ligand conjugation can also be performed using click chemistry, which is designed to run in mild reaction conditions. The most extensively exploited reaction is the copper catalysed azide-alkyne cycloaddition [154]. The by-products of the reaction, 1,3-triazoles, are biocompatible and approved to be used in drug formulation. Strong binding affinities between biomolecules can be harnessed to ensure that ligands such as antibodies are in correct orientation. The biotin–avidin interaction is potent and has been employed to improve anti-EGFR targeting in breast cancer therapy [155].

## 7. Chemical Modifications

### 7.1. Modification to Nanocarriers

The properties of nanocarriers, such as their physical and chemical stability, similarity of size to biomolecules and efficiency for drug-loading, make them a promising therapeutic tool for gene delivery [156].

The delivery of drugs with nanocarriers increases the therapeutic index. However, to have the desired effects, the therapeutic must avoid clearance and reach the target tissues in appreciable quantities. As such, most nanocarriers require surface coatings to reduce immune clearance, prolonging the circulation time and allowing the carriers to reach their target site (Table 4) [157]. The most widely used surface coating is polyethylene glycol (PEG), a hydrophilic polymer initially known to have very little immunogenicity. PEG creates a hydrophilic layer on the nanocarrier surface to reduce uptake by the phagocytic cells by preventing interactions with the receptors of phagocytic cells [158]. This decreases immune clearance and leads to increased circulatory half-life and bioavailability. However, a recent challenge with PEGylation is the development of anti-PEG antibodies, which may be a consequence of constant exposure to products that contain the polymer [159].

Bio-inspired shielding strategies may be used to circumvent this challenge. These strategies rely on the use of biodegradable biomolecules including lipids, proteins and carbohydrates to actively stealth the nanoparticles in the body. All cells in the human body are surrounded by a lipid membrane. This can be mimicked in therapy by enveloping the nanocarriers in lipid membranes derived from cells in the body. The enveloped NCs would avoid clearance by the immune system as they would be recognised as an endogenous material and left in circulation. Increased blood retention times were observed with the biomimetic coating with red blood cells as a result of decreased phagocytosis of nanoparticles. Repeated administration does not result in toxicity or accelerated blood clearance in vivo [160]. Proteins can be utilised in a similar fashion. Nanocarriers can be coated with “self” membrane glycoproteins or proteins with high abundance in the body such as CD47 and serum albumin. Carbohydrates that interact with amino acids in the body, such as sialic acid, and those that are normally conjugated to cell surface proteins, such as glycosamininoglycans, make good shielding candidates. They have limited toxicity and immunogenicity and can be targeted at certain carbohydrate-binding receptors [157].

The main goal of drug release kinetics control is to maintain the drug level in blood within the therapeutic window. Physicochemical changes to the structure of the nanoparticles, making them responsive to external stimuli such as changes in pH, heat and light, can result in drug release at a particular time and physiological location. For drug delivery to infected and inflamed tissues or tumour sites with acidic pHs, nanocarriers can be modified to carry acidic or basic groups extending from the surface that accept or release protons in response to pH changes at the target site. In an example of this, a pH-sensitive liposome system was prepared by modifying the surface of liposomes with 3-methylglutarylated poly(glycidol) (MGlu-PG) [161], and the pH sensitivity came from carboxylates on the MGlu-termini, which destabilized liposomes as they were protonated in acidic pH. Silica NPs can be formulated to have tuneable pore sizes that will confine drugs in their cage structure for sustained release [162]. The surface of silica NPs can be modified with amine groups that establish ionic interactions with the drug being carried to delay their release [163]. To prevent premature release of the content, silica NPs can be capped with compounds such as calcium phosphate (CaP) precipitates to block the pores [164].

### 7.2. Modification to siRNA

The modifications discussed so far have been targeted at the nanocarriers, however, the structure of the siRNA molecule can also be modified, as reviewed by Hu et al. [165]. Such modifications may enhance stability and improve the potency of siRNA while lowering the risk of toxicity. The ribose sugar, the phosphate linkage of the phosphodiester bond, and the bases are the sites for siRNA modification. Ribonucleases require the 2′-OH group on the ribose for the hydrolysis of RNA. Thus, modifying the groups at the 2′ position prevents the ribonucleases from cleaving the RNA molecule. The most frequently used modification is 2′-O-methyl (2′-OMe), which, in addition to protecting the siRNA from nucleases, has been shown to increase its affinity for the target mRNA [166]. Another common modification is 2′-deoxy-2′-fluoro (2′-F). Analogues of these, in modifications such as 2′-O-methoxyethyl and 2′-arabino-fluoro, have been developed to further increase affinity for the target [167]. Modifications have also been made in the ribose ring to produce unlocked nucleic acids (UNA) and glycol nucleic acids (GNA). Incorporation of these into the siRNA molecule results in thermal destabilization, which increases the RNAi activity and reduces off-target effects [165].

The use of base replacement in the production of nucleic acid-based drugs is extremely beneficial. The use of base analogues in the sequence of the siRNA may reduce immune recognition and also reduce susceptibility to nucleases [165]. Substitutions at key positions on the naturally occurring base can also be utilised to achieve increased RNAi activity. An example of this was observed in the suppression of targeted gene expression in tumour cells with siRNAs containing 5-fluoro-2′-deoxyuridine (FdU) substitutions [168]. Another site for modification on the siRNA molecule is the phosphonate group. Replacement of one of the nonbridging oxygens of a phosphodiester with sulphur creates a phosphorothioate (PS) linkage which has been shown to enhance nuclease resistance and increase binding to plasma proteins, resulting in prolonged circulation time. The number of PS linkages present in the siRNA, their positions and conformation affect the RNAi activity [165,169].

## 8. Progression of RNA Therapeutics to the Clinic: Manufacturing Challenges

The fabrication and manufacture of siRNAs and nanocarriers is a complex process and has been a major bottleneck in getting the treatments to the clinic. This is mainly due to a lack of reproducibility and scalability. However, a newer platform, the NanoAssemblr^®^, has addressed these issues and revolutionised the process, particularly for lipid nanoparticles. The platform is based on Precision Nanosystems’s NxGen microfluidic technology for producing consistent size, payload encapsulation and biological activity between batches [170]. The platform enables reproducibility, quality and batch control, in addition to scalability and faster production times. Four systems are currently available, depending on the scale and application. Spark^®^ enables small-scale preparations (50–200 µL) of the LNPs in the laboratory, useful for screening nucleic acids and LNPs and also for drug discovery and validation [171]. The Ignite^®^ platform is beneficial for optimizing NP formulation for preclinical studies by enabling very controlled, precise and accelerated assembly of LNPs between 1–20 mL. The Blaze^®^ has high capacity and can easily scale up formulations developed on the Ignite^®^ up to 10 L per formulation. A GMP System is also offered, with a simple workflow for preparing formulations ready for clinical, commercial use [171].

## 9. Conclusions and Future Perspective

Sustainable and equitable, long-term HIV control, without the need for ART, remains a challenge. The development and use of gene therapy, particularly RNA interference, in the successful treatment of other medical conditions, together with the preliminary results from research involving HIV, provide evidence for their potential as a tool for HIV control. Nanocarriers serve as an excellent platform for the safe and efficient delivery of RNA therapeutics in the human body, with extensive research underway to achieve more effective ways of improving the safety, bioavailability and specificity of active targeted delivery. We envisage that the nanotechnology field will continue to evolve and offer novel platforms with increased functionality of therapeutics to enhance treatment of HIV and other diseases.

## Figures and Tables

**Figure 1 pharmaceutics-14-01352-f001:**
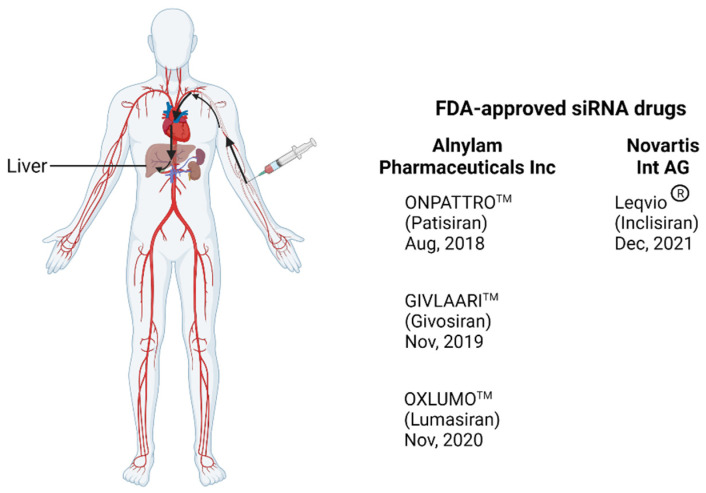
Illustration of the delivery route of FDA-approved siRNA from injection to the liver through the circulation system. The drug enters the circulatory system after it is administered intravenously and reaches the liver where it targets hepatocytes. Created with BioRender.com.

**Figure 2 pharmaceutics-14-01352-f002:**
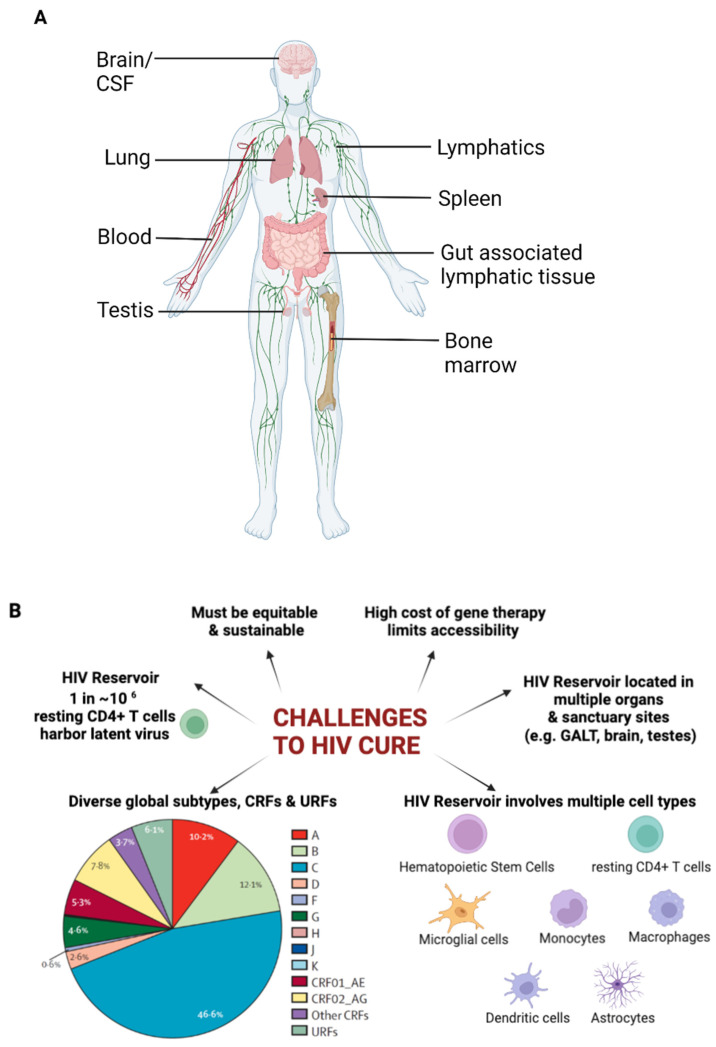
Challenges associated with HIV infection. (**A**) Illustration showing the distribution of organs and tissues known to harbour latent HIV (**B**) Challenges in attaining HIV cure. Created with BioRender.com.

**Figure 3 pharmaceutics-14-01352-f003:**
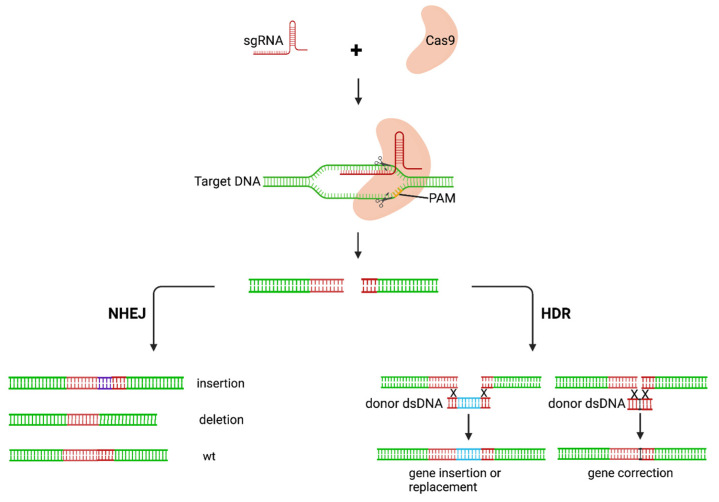
Mechanism of action of CRISPR-Cas9 gene editing. The CRISPR-Cas9 complex cleaves the target and creates a DSB. The DSB is then repaired via either the NHEJ or HDR pathway. The NHEJ pathway is error-prone, resulting in loss of gene function caused by indels in gene sequence. HDR results in precise modifications of the target. sgRNA, single guide RNA; PAM, protospacer-adjacent motif; NHEJ, nonhomologous end-joining; HDR, homology-directed repair; wt, wildtype. Created with BioRender.com.

**Figure 4 pharmaceutics-14-01352-f004:**
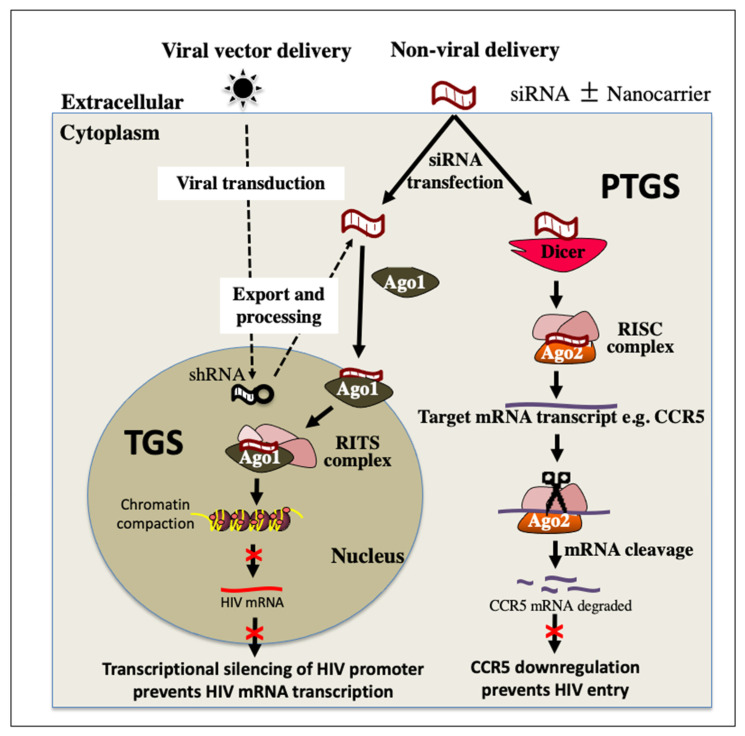
Proposed combination of PTGS and TGS RNAi pathways to control HIV infection. Both RNAi pathways can be mediated by viral or non-viral delivery of RNA sequences. siRNA can achieve PTGS via RISC initiating specific cleavage of mRNA transcripts of cellular factors required for HIV infection or replication such as CCR5. A downregulation of CCR5 prevents entry of HIV into the cell. siRNA can also trigger TGS in the nucleus via the RITS complex. This initiates repressive epigenetic modifications, such as increased histone methylation and deacetylation, at the HIV promoter region, resulting in transcriptional silencing and abrogation of the replication cycle. PTGS, post-transcriptional gene silencing; TGS, transcriptional gene silencing; Ago1, Argonaute 1; Ago2, Argonaute 2; shRNA, short hairpin RNA; siRNA, small interfering RNA; RISC, RNA induced silencing complex; RITS, RNA induced transcriptional silencing complex; mRNA messenger RNA; CR5, C-C Motif Chemokine Receptor 5.

**Figure 5 pharmaceutics-14-01352-f005:**
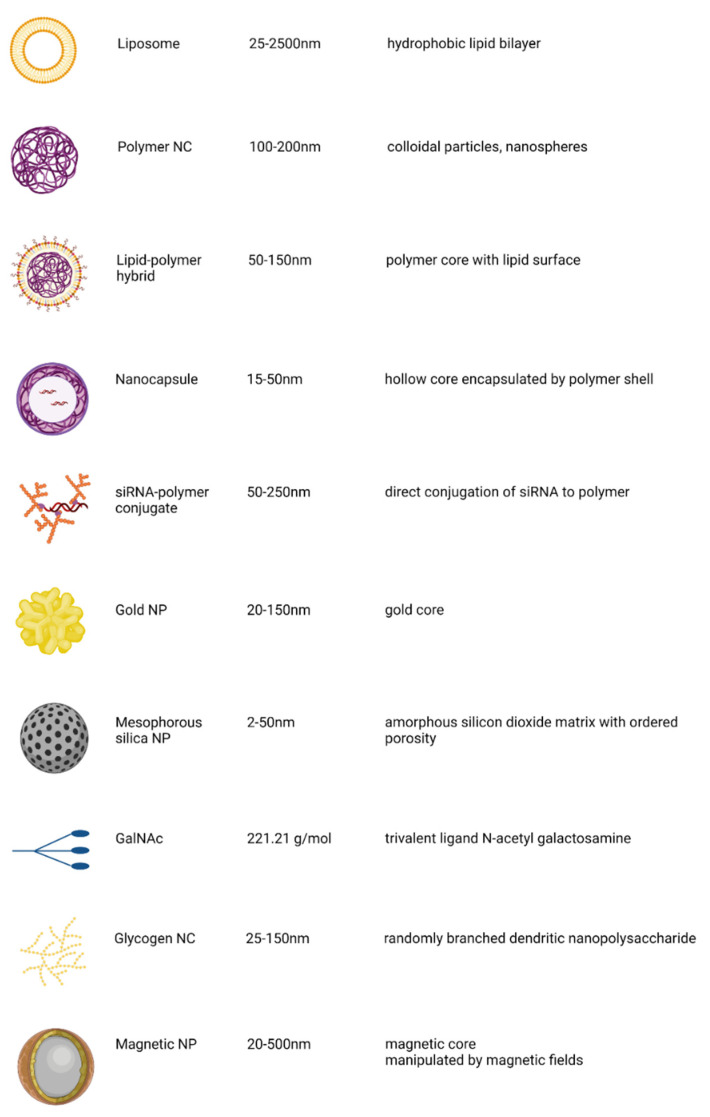
Summary of non-viral delivery systems currently used for nucleic acid delivery. Created with BioRender.com.

**Figure 6 pharmaceutics-14-01352-f006:**
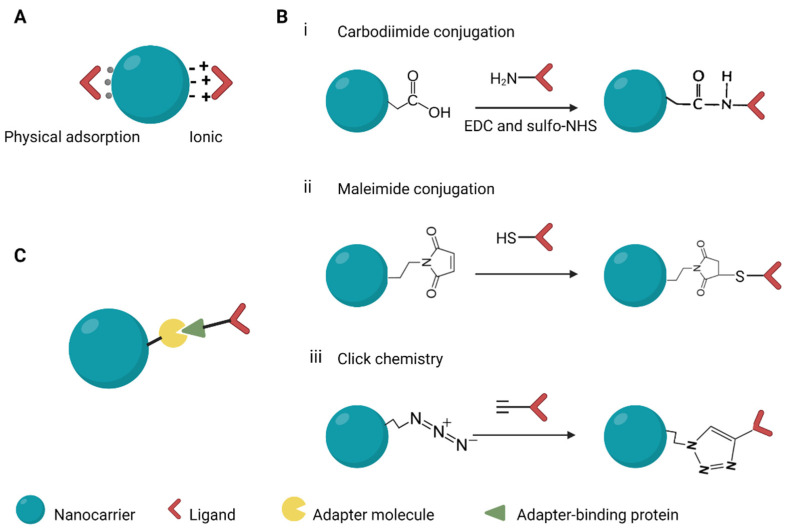
Methods used in conjugating ligands to nanocarriers. (**A**) Adsorption, (**B**) covalent conjugation, (**C**) use of adapters. EDC, 1-ethyl-3-(-3-dimethylaminopropyl) carbodiimide; sulfo-NHS, N-hydroxysulfosuccinimide. Created with BioRender.com.

**Table 1 pharmaceutics-14-01352-t001:** Selected non-viral systems used for delivery of RNA therapeutics in clinical trials and pre-clinical development.

Non-Viral System	Nucleic Acid	Target	Condition	Clinical Trial Stage	Current Clinical Trials.gov Identifiers	Reference
*Lipid nanoparticles*						
Liposome	Tetravalent RNA drug products	selected malignant melanoma-associated antigens	Melanoma	I	NCT02410733	[76]
Lipid NP	siRNA	Polo-like kinase 1	Hepatocellular carcinoma	I/II	NCT02191878 complete	[77]
Lipid NP	siRNAs	*VEGF* and kinesin spindle protein	Pancreatic Ductal Adenocarcinoma, Pancreatic Cancer	II	NCT01158079 NCT00882180	[78]
Liposome	siRNA	EphA2	Solid tumours	I	NCT01591356	[79]
Amphoteric liposomes	dsRNA	CEBPA gene	Advanced liver cancer and solid tumours	I	NCT02716012 NCT04105335	[80,81]
Lipid NP conjugate to vitamin A	siRNA	HSp47	Hepatic and pulmonary fibrosis	II	NCT03538301	[82]
Proprietary lipid NP technology	2 mRNAs	encode heavy and light chains of anti-Chikungunya antibody	Chikungunya infections	I	NCT03829384	[83]
Lipid particle	siRNA	Three viral genes	Hepatitis B	II	NCT02631096	[84]
Lipid NP	mRNA	Encode anti-CMV antibodies	Cytomegalovirus vaccine	II	NCT04232280	[85]
Lipid NP	mRNA	Encode anti-hMPV and PIV3 antibodies	human metapneumovirus and parainfluenza virus type 3	I	NCT04144348	[86]
Lipid NP	mRNA	Encoding a prefusion F glycoprotein	Respiratory Syncytial Virus	I	NCT04528719	[87]
Lipid NP	mRNA	cystic fibrosis transmembrane conductance regulator	Cystic fibrosis	I/II	NCT03375047	[88,89]
*Inorganic nanoparticles*						
Gold NP	siRNA	Bcl2L12 mRNA	Glioblastoma	I	NCT03020017	[90]
SPION	siRNA	HIV nef	HIV	-		[91]
*Polymeric*						
siRNA-polymer bioconjugates	siRNA	RRM2	Solid Tumor Cancers	I	NCT00689065 complete	[92]
Polymeric complexes	siRNA	KRAS G12D	Pancreatic ductal adenocarcinoma, pancreatic cancer	II	NCT01676259	[93,94]
Chitosan NP	2 siRNAs	SART3 and hCycT1	HIV	-		[95]
Nanocapsule	siRNA	CCR5	HIV	Preclinical		[96]
*Nanocells*						
Targeted nonliving bacterial minicells	miRNA	miR-16-based mimic	Malignant Pleural Mesothelioma, NSCLC	I	NCT02369198 complete	[97]
*GalNAc conjugation*						
GalNAc	siRNA	RNAi therapeutic targeting transthyretin (vutrisiran)	Amyloidosis	III	NCT03759379	[98,99]
ESC-GalNAc	siRNA	hepatic expression of *LDHA* (nedosiran)	primary hyperoxaluria	I and II	NCT03392896 NCT04580420	[100,101]
ESC-GalNAc	siRNA	antithrombin (fitusiran)	haemophilia A and B	II and III	NCT03417245 NCT03417102 NCT03974113	[8]
Naked siRNA	siRNA	p53 mRNA (teprasiran)	prophylactic treatment for acute kidney injury (AKI) following kidney transplant or cardiovascular surgery	II/III	NCT02610283 NCT03510897 NCT02610296	[8,102]

**Table 2 pharmaceutics-14-01352-t002:** Summary of specific disease types to demonstrate the wide range of specific organ and cell targets. There is increasing difficulty of targeting therapeutics for disease treatment when a greater number of affected cell types are involved.

Disease	Cell Type	Location
Hepatitis B	Hepatocytes	Liver
COVID-19 disease	Epithelial cells Alveolar type II cells	Nasal cavity (upper respiratory tract) Lungs (lower respiratory tract)
Cystic fibrosis	Mucoid-producing cells	Lungs Pancreas Intestines
HIV	CD4+ T cells Myeloid cells Astrocytes Epithelial cells Microglia	Lymph nodes Gut associated lymphoid tissue Reproductive tissue Brain Lungs

**Table 4 pharmaceutics-14-01352-t004:** Examples of chemical modification of nanocarriers and siRNAs.

Target Molecule	Modification
Nanocarrier	PEGylation
	Shielding with biodegradable biomolecules
	Physicochemical changes to increase sensitivity to external stimuli such as changes in pH, heat and light
siRNA	2′ OH modification such as 2′-OMe and 2′-F
	Thermal destabilization with UNA and GNA
	Modification of bases and analogue base substitutions
	Phosphorothioate linkage
	Bioconjugation

## Data Availability

Not applicable.

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
