# Peer review of "Targeted Nanocarrier Delivery of RNA Therapeutics to Control HIV Infection"

_pharmaceutics, 2022, doi:10.3390/pharmaceutics14071352_

Round 1

Reviewer 1 Report

The manuscript by Agbosu and colleagues reviews the current findings of RNA therapeutics to control HIV-1 infections.  The authors discuss various problems associated with a curing of HIV-1 from infected patients which include the cell types involved and  the location of HIV-1 reservoirs. The authors also discuss the cure strategies that have been employed including hematopoietic stem cell transplants, CRISPR/Cas editing of HIV-1 genomes, shock and kill and block and lock. This followed by a discussion on RNA interference strategies and the delivery methods that are being employed to deliver the RNAi to cells including nanocarriers, lipid nanoparticles, inorganic nanocarriers, polymer-based systems, and GalNAc conjugation.  While these delivery methods are in their infancy and are associated with various biological problems such rapid clearance, the review provides an interesting account how these problems are being investigated and potential future strategies. Overall, the manuscript is a comprehensive review on the subject with only a few grammatical errors, which I have listed below.   

1. line 14: fatal disease to a chronic….. Suggest replacing in to “into.”

2. Line 71: “reaches the hepatocyte where it targets…” Suggest changing to “reaches the liver where it targets hepatocytes…”

3. Line 109-110: “involves” should be changed to “involved

4. line 366:The vaccines used against SARS-CoV-2 are not yet “approved” but have been given emergency use authorization.

5. Line 537: the word accumulate should be changed to “accumulation.”

6. Line 629: “deliver si-RNA should be changed to “deliver siRNA.” The hyphen is unnecessary.

7. Line 691: “apH-sensitive” should be changed to “a pH-sensitive

Author Response

We thank Reviewer 1 for their excellent suggestions and have incorporated all of them into the manuscript.

Reviewer #1:

Comments and Suggestions for Authors

  1. line 14: fatal disease to a chronic….. Suggest replacing in to “into.”

Response: The change has been effected on line 13.

  1. Line 71: “reaches the hepatocyte where it targets…” Suggest changing to “reaches the liver where it targets hepatocytes…”

Response: The sentence in line 87 and 88 has been altered to read “reaches the liver where it targets hepatocytes”

  1. Line 109-110: “involves” should be changed to “involved”

Response: Involves has been changed to involved on line 132.

  1. line 366:The vaccines used against SARS-CoV-2 are not yet “approved” but have been given emergency use authorization.

Response: The Pfizer-BioNTech and Moderna COVID-19 Vaccines were approved by the US FDA in August 2021 and January 2022, respectively.

  1. Line 537: the word accumulate should be changed to “accumulation.”

Response: The correction has been made on line 624.

  1. Line 629: “deliver si-RNA should be changed to “deliver siRNA.” The hyphen is unnecessary.

Response: The correction has been made on line 728.

  1. Line 691: “apH-sensitive” should be changed to “a pH-sensitive”

Response: The correction has been made to line 831.

Reviewer 2 Report

In this manuscript, the authors reviewed the Targeted nanocarrier delivery of RNA therapeutics to control HIV infection. In my opinion, some issues should be further addressed and I hope the following comments could be helpful for improving their paper.
  1. In the introduction, the background about  Targeted nanocarrier for delivery of RNA therapeutics is little, the authors should enrich this part and emphasize the necessity of "targeted nanocarrier delivery of RNA therapeutics " for HIV infection control. 
  2. Authors focused on HIV infection control, but what are the distinguished properties and specific problems of HIV infection? The authors never discussed it.
  3. According to the applications of Targeted nanocarrier delivery of RNA therapeutics, most, if not all are applicable for other kinds of diseases. Then why did the authors not expand the topic to other disease therapy?
  4. Good quality figures are very important for a good review paper, kindly improve the quality of the figures. Try to add at least 4-5 more figures in this manuscript from the recent literature. 
  5. The authors should summarize the current approaches of fabricating "Targeted nanocarrier for delivery of RNA therapeutics" and compare their advantages and disadvantages in order to draw the reader's attention.
  6. This manuscript is well organized but lacks specific comparative analysis. What are the advantages of "Targeted nanocarrier for delivery of RNA therapeutics" compared with traditional technology?
  7. Please revisit the entire manuscript for minor grammar issues.
  8. Future perspective is missing in this manuscript kindly add it in this manuscript briefly.
  9. In conclusion, the author should consider giving some methodological design about how to improve the performance of the Targeted nanocarrier.

Author Response

We thank Reviewer 2 for their insightful comments and suggestions and have incorporated all of them into the manuscript as described below.

Reviewer #2:

Comments and Suggestions for Authors

  1. In the introduction, the background about Targeted nanocarrier for delivery of RNA therapeutics is little, the authors should enrich this part and emphasize the necessity of "targeted nanocarrier delivery of RNA therapeutics " for HIV infection control.

Response: The following sentences have been added to the introduction. “Due to this, it may be necessary to use targeted nanocarriers for effective delivery of gene cargoes including RNA therapeutics. Approaches to target nanocarriers to specific sites are extensive; some of which are described in this review. For efficient control of HIV, therapy would need to be targeted at multiple cells types (involved in the latent reservoir) and at multiple sites, as shown in Figure 2. Thus, a combination of different targeting approaches may be required” (lines 74 – 79)

  1. Authors focused on HIV infection control, but what are the distinguished properties and specific problems of HIV infection? The authors never discussed it.

Response:Additional sentences (lines 69-71 and 79-82) and a figure (Figure 2b, line 90-92) on the challenges associated with HIV infection have been added.

  1. According to the applications of Targeted nanocarrier delivery of RNA therapeutics, most, if not all are applicable for other kinds of diseases. Then why did the authors not expand the topic to other disease therapy?

Response: We absolutely agree that targeted nanocarrier delivery can be applied to other diseases where gene silencing is required, although we feel it is outside the scope of this review to be able to comprehensively cover all possible diseases that may require targeted nanocarriers for effective therapy. However, in section 6 we have included the sentence on line 615 “RNA therapeutics have broad ranging application for many diseases that would be curable by gene silencing”. to illustrate that it is a feasible approach for other diseases, and highlighted two other disease in comparison to HIV in Table 2. We indicated that HIV, being multisytemic infection, was more challenging to tackle and that is why we focused the review on HIV control.  

  1. Good quality figures are very important for a good review paper, kindly improve the quality of the figures. Try to add at least 4-5 more figures in this manuscript from the recent literature.

Response:                                                                                                                                                                                                                   We absolutely agree and have improved the quality, as well as adding two additional figures; Figure 2b and Figure 6 on lines 90-92 and 788, respectively.

  1. The authors should summarize the current approaches of fabricating "Targeted nanocarrier for delivery of RNA therapeutics" and compare their advantages and disadvantages in order to draw the reader's attention.

Response: To incorporate the reviewer’s excellent suggestion, we have added a section on the manufacturing challenges of nanocarriers and RNA therapeutics (lines 872-889).

  1. This manuscript is well organized but lacks specific comparative analysis. What are the advantages of "Targeted nanocarrier for delivery of RNA therapeutics" compared with traditional technology?

Response: Thank you for identifying this point. We described the traditional technology of viral vectors in section 5, and have now incorporated the comparison of passive (i.e. non-targeted) delivery with active (ligand targeted) delivery to clarify targeted delivery with “traditional” nanocarrier platforms. This appears in the text on lines 387, 410, 586, 620, 626, 65, 691, 693.

  1. Please revisit the entire manuscript for minor grammar issues.

Response: The manuscript has been edited for grammatical errors.

  1. Future perspective is missing in this manuscript kindly add it in this manuscript briefly.

Response: We have incorporated the future perspective into the conclusion, along with the active targeted delivery point and changed the subheading to read “Conclusions and future perspective”

  1. In conclusion, the author should consider giving some methodological design about how to improve the performance of the Targeted nanocarrier.

Response: We thank the reviewer for raising this important point and have included a  new section (6.2) to specifically describe the methodologies for target ligand conjugation to NCs (lines 757-783).
